# Rapid Prototyping of Virtual Reality Cognitive Exercises in a Tele-Rehabilitation Context

**Damiano Perri** [1,*] , **Martina Fortunelli** [2,*] , **Marco Simonetti** [1] , **Riccardo Magni** [2] , **Jessica Carloni** [2] and **Osvaldo Gervasi** [3]

1   Department of Mathematics and Computer Science, University of Florence, 50134 Florence, Italy; m.simonetti@unifi.it
2   Pragma Engineering Srl, 06135 Perugia Italy; riccardo.magni@pragmaeng.it (R.M.); jessica.carloni@libero.it (J.C.)
3   Department of Mathematics and Computer Science, University of Perugia, 06123 Perugia, Italy; osvaldo.gervasi@unipg.it
*   Correspondence: damiano.perri@unifi.it (D.P.); martina.fortunelli@gmail.com (M.F.)

**Abstract:** In recent years, the need to contain healthcare costs due to the growing public debt of many countries, combined with the need to reduce costly travel by patients unable to move autonomously, have captured the attention of public administrators towards tele-rehabilitation. This trend has been consolidated overwhelmingly following the Covid-19 pandemic, which has made it precarious, difficult and even dangerous for patients to access hospital facilities. We present an approach based on the rapid prototyping of virtual reality, cognitive tele-rehabilitation exercises, which reinforce the group of exercises available in the Nu!reha platform. Patients who experienced injury or pathology need to practice continuous training in order to recover functional abilities, and the therapist needs to monitor the outcomes of such practices. The group of new exercises based on the rapid prototyping approach, become crucial especially in this pandemic period. The Virtual Reality exercises are designed on Unity 3D to empower the therapist to set up personalized exercises in an easy way, enabling the patient to receive personalized stimuli, which are essential for a positive outcome in the practice. Furthermore, the reaction speed of the system is of fundamental importance, as the temporal evolution of the scene must proceed parallel to the patient's movements, to ensure an effective and efficient therapeutic response. So, we optimized the virtual reality application in order to make the loading phase and the startup phase as fast as possible and we have tested the results obtained with many devices: in particular computers and smartphones with different operating systems and hardware. The implemented method powers up the Nu!Reha system®, a collection of tele-rehabilitation services that helps patients to recover cognitive and functional capabilities.

**Keywords:** Blender; cognitive exercises; Nu!Reha; tele-rehabilitation; Unity3D; virtual reality

## 1. Introduction

Prototyping is a tool that lets anyone be able to create a so-called prototype, i.e., the basis used to make new products. Rapid prototyping is a faster prototyping technique, that allows us to reduce production costs, and it is best known as Rapid Application Development (RAD). This technique involves the use of CASE (Computer-Aided Software Engineering) tools and implies software development through graphical and visual interfaces and libraries.

In general, RAD approaches to software development privileges the adaptive process rather than planning. It is characterized by better flexibility against vague specifications or not already closed-loop development, a general risk reduction against the rigidity of plan-driven waterfall design. On the other hand, rapid prototyping idea is often linked to agile production concepts [1,2]; these gain importance in recent years as an immediate and easy answer to more and more sophisticated customer expectations.

The attention to bring services operated remotely has grown in the last years. The pressure on health-related services prompted for simple and fast solutions for tele-rehabilitation in several fields. Those services were conceived as reliable and routinely services (not only pilot tests) [3], and are nowadays even more important to provide continuous and efficient health services.

Cognitive rehabilitation is a branch of all rehabilitative treatments, and is devoted to all patients who need continuous training of cognitive functions because of injury (Traumatic Brain Injury, Stroke, Cerebral Palsy, etc) or pathology (Alzheimer's disease, Multiple Sclerosis, etc), which are affecting the Central Nervous System (CNS).

In general terms, to tune the subsequent therapy phases, results from cognitive routinely exercises must be logged and tracked by the therapist. She/he can monitor the execution of all activities, which can be either based on paper and pencil or software tools.

Pilot cognitive tele-rehabilitation practices are reported in literature: they report relevant advances for patients and their families, either objective results or perceived comfort, even if some barriers, because of remote treatment modality, have been identified as well.

A severe critical aspect is raised when high-performing hardware devices are requested because of their huge costs; so the health institution should help by giving them to several patients and caregivers, but the reality is a little different. In most cases, a limited number of the available standardized exercises reduces the motivation of neurological patients to continue the practice. Furthermore, these exercises lack contextual personalized stimuli and seem to limit personal motivation, inhibiting long term application to the proposed tele-treatment.

On the basis of these experiences, new concepts have recently become established. First of all, the opportunity and the importance of using most of the patient's devices and networks. Furthermore, it is important how to choose, among the proposed activities, the most suitable ones to improve the rehabilitation practice. Finally, we ought to highlight the importance of these activities' customization by using cognitive contextual stimuli, so that the patient's motivation should be reinforced.

The NU!Reha (NU!Reha is a Trademark project of Pragma Engineering srl, Italy) system [4] has been designed to support cognitive functions and to allow user–engaging activities. The system has been extended to let the patient carry out more complex, game-based and attractive experiences, implemented in a virtual reality environment.

Since several software components may be reused in different exercises, we provided a number of pre-programmed software elements (assets) to speed-up the development of new virtual reality exercises. In this way, we could turn this tool, based on a rapid prototyping approach, into a powerful Rapid Application Development (RAD) system devoted to the realization of cognitive exercise.

## 2. Related Works

In terms of educational resources, several solutions have been developed starting from authoring tools produced for educational purposes: presentation, interactive exercises, algorithm exploration, problem-solving activities and, more recently, pet-bots (including also Lego robotics) control for social educational games. The approach to small app generation through a graphical syntax rule control [5] made programming easier by the prevalence of block programming solutions. This winning idea [6] was quickly accepted by a large community and using similar solutions has become very common for developing single applications thought for different OSs [7].

It is fundamental to cite an important solution for industrial automation and control, to design virtual instrument solutions, such as Labview Graphical Programming environment [8].

All solutions, coming from the educational and industrial environment, had been usefully applied to user interface adaptations/configuration as well as for specific applications in AT (Assistive Technology) solutions. ICF (International Classification of Functioning,

Disability and Health) [9] draws up a rank about several human functional profiles, to be able to ensure a quick response to the needs of users by RAD solutions.

Tele-rehabilitation is a method to bring rehabilitation services to break down barriers of distance, time, and cost [10]. Documents on tele-rehabilitation first appeared in 1959, but the word was born in 1997 when the Department of Education's National Institute on Disability and Rehabilitation Research of United States proposed the field of tele-rehabilitation for the new rehabilitation engineering research center [11].

Tele-rehabilitation has been developed to improve and optimize the rehabilitation services and patient's outcome, and let her/him keep on rehabilitation path at home. The idea is to help and reinforce traditional services by using innovative technologies.

Tele-rehabilitation was first introduced in motor rehabilitation; where its usefulness in cognitive rehabilitation was discovered as well. The impairment of certain cognitive abilities can strongly depend on a number of factors and circumstances, most of them are age-related problems, such as dementia, vascular problems, head injuries, chronic psychiatric illnesses or internist pathology. After a careful and deep neuro-psychological evaluation that shows the presence of cognitive deficits, it is possible to carry out neuro-psychological rehabilitation cycles to improve cognitive functioning, stabilize deficits, stimulate residual cognitive abilities and slow down the course of decay.

Evidence shows that the exposure to a stimulus-rich environment, through learning and cognitive stimulation, can promote brain plasticity and yield recovery after brain injury or slow down the progression of a degenerative disease [12].

Tele-rehabilitation technologies can be classified as:

- Image-based technologies, which have been used in videoconferencing or for monitoring and diagnosis purposes.
- Sensor-based technologies, which use tilt switches, accelerometers or gyroscopes to sample and measure movements in a three-dimensional space.
- Virtual environments that use virtual reality to elicit specific movements from the patient.

Virtual reality technologies are useful in rehabilitation because it can generate virtual environments and implement activity repetition, feedback and motivation in order to stimulate learning of impaired skills and then translate them into the real world [13]. Virtual Reality based tele-rehabilitation systems include those using video games [14–16].

According to Willer and Corrigan [17], the recovery of motor, learning or cognitive functions in environments familiar to the patient, is a key ingredient to maximizing the results of good rehabilitation therapy. Virtual reality technologies play a crucial role towards this recovery process, enabling the therapist to reconstruct an immersive virtual scene where the patient can practice, and which is very close to the home environment.

We may assert that virtual reality has made tele-rehabilitation easier and widely adopted because of its low cost and the broad diffusion of all the necessary instrumentation. Moreover, through the game-like structure, it is possible to increase the therapeutic work in a playful context without giving up on obtaining the important curative results.

### 3. NU!Reha Platform

NU!Reha is the general framework for the occupational rehabilitation desk system, composed of two different parts: the NU!Reha Desk (a group of physical objects used by the patient to regain neurological abilities) and NU!Reha Service (a digital platform for cognitive rehabilitation).

The studies described in the previous section are the foundation of NU!Reha platform and give a scientific basis to the effectiveness of performing cognitive exercises. What we mean to introduce as an innovative element is the use of rapid prototyping to develop new cognitive exercises that can be added to the platform exercises' set. This has been done by developing assets in Unity that facilitate and speed up the creation of new exercises. The chosen assets are the more recurrent elements for rehabilitation exercises.

NU!Reha Service allows to train cognitive functions such as memory, attention and executive function, changing the difficulty of activities according to the patient's abilities. This characteristic determines a flexibility service that allows a customization of rehabilitation path [4,18–26]. This flexibility implies a variety of proposed activities, and therapists can improve the platform by making new personalized exercises. The target users can be divided into two categories of patients:

- Neurological, in the context of brain injuries, with the aim to recover impaired cognitive skills and/or enhancing residual ones.
- Geriatric, not only for pathological but even physiological aging, in order to preserve the functions.

Based on the functional profile of the individual patient, the therapist can choose among activities that go to train:

- Attention: its role is to select, filter and activate relevant information for the goal, choose necessary activities and discard the useless ones that would only end up saturating the cognitive system and waste efficiency. Specifically, you can find activities aimed at:
  - Selective attention includes all those processes that involve the possibility of ignoring meaningless information and allow you to select only the necessary one, focusing on it.
  - Sustained Attention allows you to pay attention for an extended period of time.
  - Divided attention is the ability to pay attention to multiple tasks and process information from two different channels.
- Attentional shifting involves alternating between two attentional focuses: you don't have to pay attention to them at the same time, but you need to switch quickly from one to the other when the task requires to do it.
- Memory is a complex function by which subject is able to acquire, preserve and reuse knowledge. In particular:
  - Short-term memory is the ability to retain for short periods of time (10–30 s.) the information just collected; it is characterized by capacity and time limits.
  - Working memory stores and processes information while performing cognitive tasks, so it allows us to keep information in memory and to manipulate it at once.
  - Long-term memory allows us to keep an indefinite amount of information for an equally indefinite time, therefore even for a lifetime.
- Executive functions are a set of related but distinct skills necessary for an intentional, targeted and problem-solving action [27]. In particular:
  - Categorization is the ability to organize information by capturing the essential characteristics, sorting and giving a meaning to our experiences.
  - Inhibition is the ability to inhibit previously learned responses and to control the interference effect of distracting stimuli.
  - Cognitive flexibility is the ability to change behavioral patterns based on received feedback.
  - Planning is the ability to imagine how to reach a goal and what steps must be taken to get to it.
  - Emotional self-regulation is the ability to recognize and manage one's emotions.

The NU!Reha service offers professionals and structures the opportunity to continue the work done in the presence even at the patient's home, in order to guarantee continuity and consolidate the results obtained. In this way, once the therapist has configured the activities to be carried out by her/his patient, she/he can monitor their progress at any time and on the basis of this, if necessary, propose a new plan. The execution of a personalized rehabilitation program at home is facilitated by the bring-your-own-device technology, which NU!Reha is based on, since a dedicated hardware system is not required, but can be run on the user's device. All this can simply be done by ensuring accessibility: indeed,

a simplified access method on a tablet has been implemented with no necessity of entering credentials, but only by focusing a QRCode by the device's camera. In addition, a group of exercises/activities have been designed to be carried out by the interface equipped with one or two sensors, so even users with poor or absent pointing or movement skills of the upper limbs can access the platform.

### 3.1. The Architecture of NU!Reha Service

The architecture of the NU!Reha system, shown in Figure 1, is composed of two parts: the back-end used by health professionals (i.e., psychologists, physiotherapists, speech therapists) who organize a patient's therapy by inserting the most appropriate exercise, and the front-end used by patients to perform the exercises.

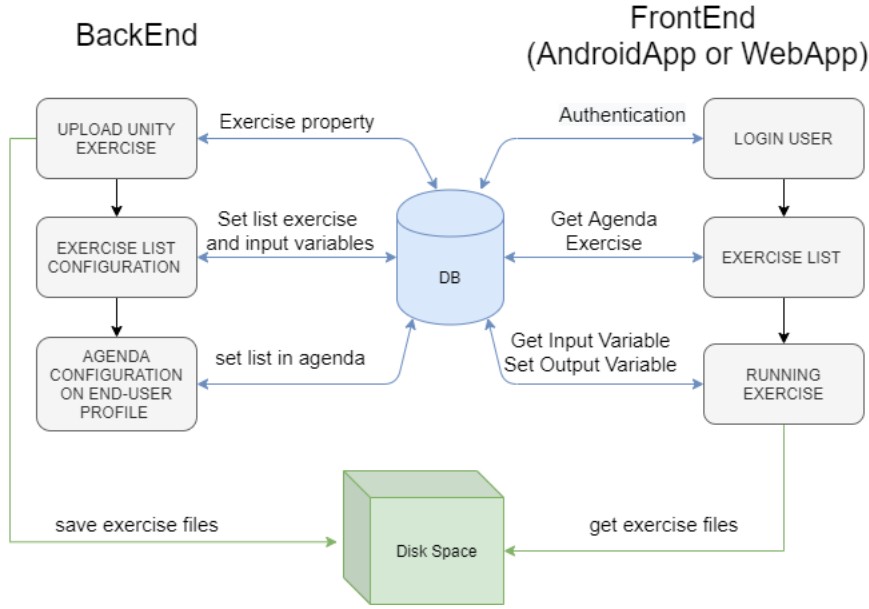

**Figure 1.** The architecture of the NU!Reha system.

The main functions of the back-end, are: upload the developed 3D exercises, list configuration composed of several exercises customized with input variables and the agenda configuration that allows us to assign to a patient an exercise list for a certain period of time. On the other hand, the main functions of the front-end are: execution of personalized exercises inserted in the agenda by the operator and variable saving that allows us to monitor the progress and results of the therapy.

In order to integrate the new VR exercises implemented with Unity 3D in the architecture described above, ad-hoc web user interfaces have been created for:

- Upload HTML exercises on cloud and save input variables in the database. The format exported by Unity is WebGL since it is fully integrated with web standards.
- Configuration and customization through input variables of previously loaded exercises to make a list.
- Execution of exercise with the possibility to manage input and output variable through database connection. All exercises use a script that manages the I/O variable, this script opens a connection to the REST server of NU!Reha via HTTP protocol. In the case of Input variables the connection gets results in JSON format, each tuple obtained is stored as an instance of a specially created class. In the case of output ones, all variables contained in exercise scripts are read and shown to the developer in the form of check-boxes to allow the selection and consequently the saving into the database through consecutive attempts in case the first one or subsequent fails.

## 4. Advanced Unity Techniques

One of the aspects that characterize this project is the creation of a virtual environment that dynamically generates a new exercise whenever it is performed. Clinical professionals told us they found it stimulating and important for the patient to face new situations anytime. In this way, the effort is to stimulate her/his cognitive functions to solve problems that are generally similar but not the same.

Unity gives us the opportunity to save our time especially in the development phase, because it features a RAD logic. It allows applications to be developed for a wide range of devices, introducing dynamic properties thanks to Assets and scripts. The term Asset refers to "objects of various kinds", i.e., animations, textures, models or even entire projects created by third parties or ourselves.

At the program start, a subset of objects gets randomly picked up from those earlier defined; for example: the book, the puppet, the camera, a wall clock, a lamp, a bottle, and so on. These objects can belong to two distinct classes: contextual and not contextual related with the scenario. Next, we have defined a series of points within the scenario, called spawn points. At the start, the software lets these objects appear in their own specific spawn point, extracted randomly among the available ones. An example of spawn points' arrangement is shown in Figure 2.

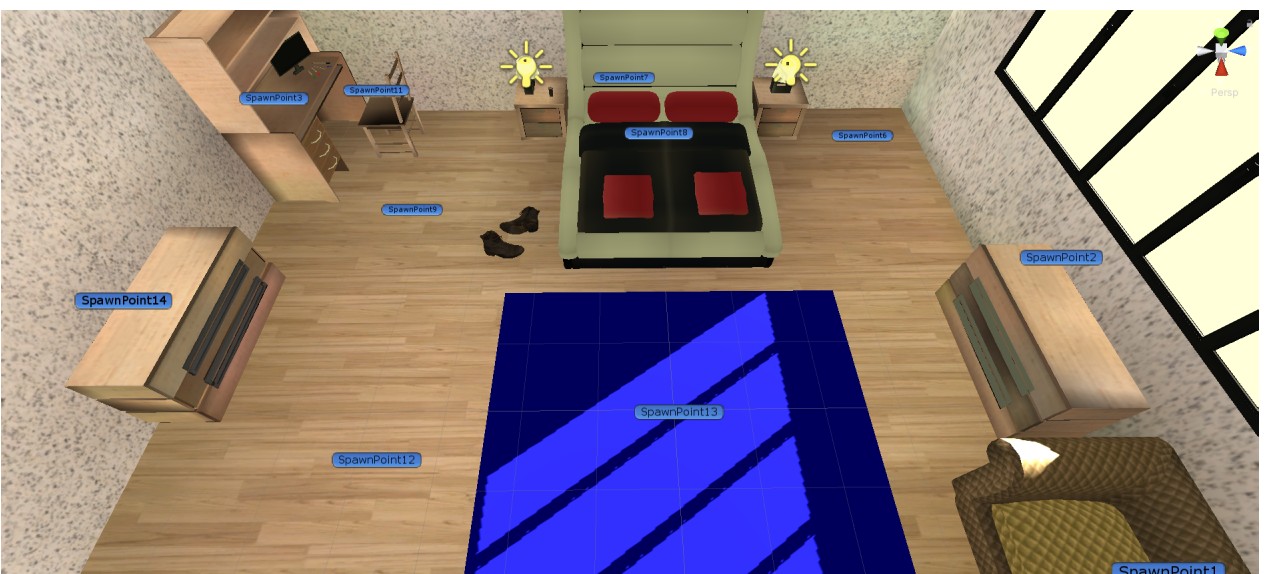

**Figure 2.** The spawn points.

In order to make the exercise easier, a simplified software version is available that helps the patient to look for an object in a list, simply by calling its name. To avoid patients learning the solutions by heart, different patterns are presented every single time the exercise is started, with different objects in different positions.

During the development of software, we took great care of all performance aspects. Our goal was to obtain the shortest application loading time to give to the users the possibility to immediately access the exercise that the therapist wants to manage. To do this we balanced various aspects, in particular, we focused on how to obtain the best compromise among graphics quality, loading time and size in megabytes when the project is compiled in WebGL. Patients who will use the software may have outdated devices and this aspect actually requires us to generate software that will require a minimum amount of RAM and VRAM while running.

Generally, outdated devices are rarely updated by smartphone manufacturers for major releases of the operating system, so we need to take advantage of APIs that allow us to obtain the highest percentage of market share even in the smartphone sector.

We have also considered that users may have slow and unstable connections, so it is important that the program should require them to download as little data as possible over the Internet.

*4.1. Graphics Settings*

To obtain the best possible performance we implemented as many technical optimizations as possible on the Unity project. First of all, we analyzed the rendering techniques. The graphic settings related to the image quality that will be shown in the rendering phase on the device was manually calibrated by creating a custom profile in the quality section of the project settings. The resolution of the textures has been set to "Full Res" thanks to the optimizations implemented with the techniques that will now be described, and there is no reason to further compress them. Anisotropic filtering and antialiasing filtering was disabled, as well as the soft particles calculation engine, since they will not be used. We have also turned off the real-time reflections. The Texture Streaming feature has been enabled. This allows us to reduce the memory quality required for textures so that only small portions of mipmaps are loaded into memory. For this functionality, we have set a maximum amount of VRAM memory that Unity can use equal to 512 MB and a maximum of 1024 simultaneous input/output requests. As for the shadows, we used the Distance Shadowmask calculation method and render only the "hard" shadows on the scene, thus ignoring the calculation of soft shadows.

This calculation method allows us to obtain a realistic and fairly faithful graphic environment. To get a little bit lighter project, we could have set the rendering in shadowmask mode, but quality loss would not have been justified by the gained performance. Indeed, the project is contained and the rendering of the room is efficiently calculated anyway. The shadow resolution was set to a medium quality level.

The Forward rendering path was set on the graphics card, which guarantees compatibility with the greatest number of devices compared to deferred mode. We describe now how the lighting system was configured, since it is well known that this is one of the aspects that most affects the rendering performance of a three-dimensional environment.

To try to contain the required computational capabilities as much as possible, we have disabled the real-time rendering of the lights, preferring instead to calculate them at compile time. To do this we deselected the RGI (Realtime Global Illumination) and we activated the "Baked Global Illumination". This operation requires that the objects should be defined with the "Static" property. All the objects that are part of the setting were therefore set as "Static" and the light bulbs that illuminate the scene were set with a "Baked" type of light. This lighting calculation methodology pre-calculates how the light will affect the objects and elements in the scene by saving the calculated information in the LightMap. In total, we defined four lights in the scene: the sun, the light in the room, the lights coming from the left bedside table and the right bedside table. The lights in the room are all Point Light set with the Baked type lighting mode. The sun is mimicked via a directional light in Mixed calculation mode. This calculation mode allows us to take advantage of the pre-calculation (baking) of the lighting that interacts with static objects but leaves the rendering engine the ability to correctly calculate the lighting of any moving objects.

The patient will find in the room several moving objects that once clicked will perform a short animation that provides the user with feedback on the correctness of the action performed. The number of Direct samples was set to 32 and the number of indirect samples to 128. Our tests showed these values are an excellent balanced set that allows us to obtain a good graphic quality while consuming few resources. Furthermore, environmental occlusion was activated to further improve the visual rendering. The Lightmaps obtained therefore has a size of $9 \times 32 \times 32$ px and occupies a space of 144 KB of memory. To minimize the space it occupies, we compressed it to obtain a size of 48.5 KB. On the player tab of the project settings panel, we needed to set the Graphics API in order to use WebGL 1.0 instead of WebGL 2.0. This operation let us further reduce the size occupied by the compiled project while maintaining a comparable graphic quality.

### 4.2. Polygonal Complexity and Texture

Then, we worked and thoroughly analyzed how to optimize the objects that make up the scene. Each single object has been processed and analyzed one by one, in order to minimize the number of polygons, finding the right balance between polygonal complexity and graphic quality. This is an operation that cannot in any way be defined a priori and strongly depends on the type of object. For example, the cushions that are placed above the bed can have very high compression, while the polygonal model of a shoe due to its shape and structure requires more definition to be appreciated on the screen.

Object compression was done using two technologies: the first is through the Blender software, the second is provided directly by Unity, which allows, via the mesh inspector, to obtain a copy of the object model that gets compressed and directly saved in the project file system. To do this we set the mesh compression and the optimize mesh for each object by calibrating the parameters one by one. Furthermore, following all these intermediate steps, we recalculated the Normals for each object by generating the individual UV LightMaps that tell the graphic engine how the light should interact with the shape in the scene.

We also paid great attention to the textures we wanted to examine and study. First of all, textures were chosen so that they could create a contrast with the background and did not overlap the colors with each other in order to improve accessibility for users.

### 4.3. Compiler Optimizations

The last optimization phase was dedicated to the final compression of the project and to the study of the techniques that allow us to reduce, as much as possible, the waiting time for the loading and start-up phase of the project. Unity provides different ways to export a project to WebGL:

- Without any kind of compression, the project has a large memory occupation and requires a considerable amount of time and bandwidth to be downloaded. Because of this difficulty, we decided not to choose this option.
- Gzip, the project is compressed using the famous algorithm designed by Jean-loup Gailly, widely used in Unix systems and published for the first time in 1992
- Brotli, a compression algorithm born in 2013 and developed by Google. This algorithm is particularly efficient when it has to compress texts.

It is also possible to choose between two ways to export the code that makes up the project:

- ASM, the "legacy" mode. The JavaScript code is optimized and exported in bytecode through the JavaScript interpreter.
- Web Assembly streaming permits us to export the code in binary format and does not require any parsing since it is ready to run. The code is also compressed to save additional space. Its execution speed is almost comparable to what we would have achieved natively using machine code.

From the preliminary tests, we observed that the best combination was WebAssembly Streaming with Brotli compression. To validate this preliminary result we analyzed and studied in-depth how these technologies affect the size of a WebGL project and its start-up time, taking into account a wide variety of hardware. With this analysis, we could precisely determine the best possible combination for our purposes.

### 4.4. The Obtained Results

In total, we tested 17 different devices among smartphones and tablets. The devices with which the software has been tested belong to different brands, have different CPU and GPU architectures and have different operating systems, including Android, Windows, Linux, and MAC Os. This wide variety of hardware has made it possible to systematically compare several kinds of the most commonly-used hardware so that we can have a clear idea of how the program behaves with the various compilation technologies and what

type of experience the end-user will get. Eight different projects, which used several combinations of technologies, was tested for every single device. These are:

- WebGL 2.0 with Gzip and Web Assembly Streaming
- WebGL 2.0 with Gzip without Web Assembly Streaming
- WebGL 2.0 with Brotli and Web Assembly Streaming
- WebGL 2.0 with Brotli without Web Assembly Streaming
- WebGL 1.0 with Gzip and Web Assembly Streaming
- WebGL 1.0 with Gzip without Web Assembly Streaming
- WebGL 1.0 with Brotli and Web Assembly Streaming
- WebGL 1.0 with Brotli without Web Assembly Streaming

We have summarized the results obtained in tables to make them easier to read. In Table 1 we carry out the analysis for WebGL 1.0 platform and we tested the combinations of Gzip or Brotli, with and without web assembly streaming.

**Table 1.** WebGL 1.0 comparison.

| *Device* | **Year** | **Operative System / Browser Web** | *WebGL 2.0 Gzip No Stream* | *WebGL 2.0 Gzip WAS* | *WebGL 2.0 Brotli No Stream* | *WebGL 2.0 Brotli WAS* |
|---|---|---|---|---|---|---|
| Asus Zenfone 5 | 2018 | Android 9 Chrome 85 | 00:20:21 | 00:17:05 | 00:14:27 | 00:16:10 |
| Amazon Fire HD 8 | 2017 | Fire OS 6 Chrome 84 | 00:20:00 | 00:20:43 | 00:21:59 | 00:12:63 |
| Samsung Galaxy Tab E | 2015 | Android 4.4.4 Chrome 81 | No webGL2.0 | No webGL2.0 | No webGL2.0 | No webGL2.0 |
| Huawei MediaPad M5 lite | 2019 | Android 8 Chrome 84 | 00:45:36 | 00:45:67 | 00:42:54 | 00:28:87 |
| Samsung Galaxy S7 | 2016 | Android 8 Chrome 84 | 00:08:99 | 00:06:52 | 00:09:26 | 00:06:07 |
| Samsung Galaxy S4 Active | 2013 | Android 5.01 Chrome 85.0.4 | No webGL2.0 | No webGL2.0 | No webGL2.0 | No webGL2.0 |
| DESKTOP CUSTOM | 2014 | Windows 10 Chrome 84 | 00:02:44 | 00:01:98 | 00:02:83 | 00:01:65 |
| Asus GL502VM | 2018 | Windows 10 Chrome 84 | 00:02:90 | 00:02:14 | 00:02:78 | 00:01:82 |
| Hp prodesk 400 g1 | 2013 | Ubuntu 18.04 Firefox 79 | 00:03:04 | 00:02:64 | 00:03:48 | 00:02:23 |
| DESKTOP CUSTOM | 2012 | Windows 10 Firefox 79 | 00:08:10 | 00:08:71 | 00:08:14 | 00:07:31 |
| Redmi Note 8 Pro | 2019 | Android 10 Chrome 84 | 00:22:21 | 00:14:34 | 00:23:07 | 00:16:54 |
| Acer Swift SF314-52 | 2018 | Windows 10 Firefox 79 | 00:15:13 | 00:14:37 | 00:14:46 | 00:12:30 |
| Honor 8 | 2016 | Android 10 Chrome | 00:19:20 | 00:19:09 | 00:21:26 | 00:15:17 |
| HP-PC ProBook 450 G6 | 2018 | Manjaro Chromium | 00:11:78 | 00:10:16 | 00:09:45 | 00:09:01 |
| ASUS H81M-D R2.0 | 2015 | Mint 20 Mozilla Firefox | 00:11:94 | 00:10:29 | 00:09:71 | 00:10:13 |
| MacBookPro 14.2 | 2017 | macOS 10.15 Safari | No webGL2.0 | No webGL2.0 | No webGL2.0 | No webGL2.0 |

In Table 2 we carry out a similar study for the WebGL 2.0 platform. In the tables we reported the project's loading times.

**Table 2.** WebGL2.0 comparison.

| Device | Year | Operative System / Browser Web | WebGL 1.0 Gzip No Stream | WebGL 1.0 Gzip WAS | WebGL 1.0 Brotli No Stream | WebGL 1.0 Brotli WAS |
|---|---|---|---|---|---|---|
| Asus Zenfone 5 | 2018 | Android 9 Chrome 85 | 00:14:72 | 00:17:92 | 00:13:57 | 00:09:51 |
| Amazon Fire HD 8 | 2017 | Fire OS 6 Chrome 84 | 00:18:75 | 00:12:51 | 00:20:07 | 00:12:62 |
| Samsung Galaxy Tab E | 2015 | Android 4.4.4 Chrome 81 | 01:31:88 | 01:05:34 | 01:35:25 | 01:01:57 |
| Huawei MediaPad M5 lite | 2019 | Android 8 Chrome 84 | 00:26:71 | 00:33:68 | 00:31:73 | 00:26:54 |
| Samsung Galaxy S7 | 2016 | Android 8 Chrome 84 | 00:09:47 | 00:06:70 | 00:08:42 | 00:05:71 |
| Samsung Galaxy S4 Active | 2013 | Android 5.01 Chrome 85.0.4 | 00:18:57 | 00:15:86 | 00:24:32 | 00:15:65 |
| DESKTOP CUSTOM | 2014 | Windows 10 Chrome 84 | 00:02:55 | 00:01:81 | 00:02:55 | 00:01:25 |
| Asus GL502VM | 2018 | Windows 10 Chrome 84 | 00:02:96 | 00:01:92 | 00:02:91 | 00:01:82 |
| Hp prodesk 400 g1 | 2013 | Ubuntu 18.04 Firefox 79 | 00:03:15 | 00:02:48 | 00:03:55 | 00:01:98 |
| DESKTOP CUSTOM | 2012 | Windows 10 Firefox 79 | 00:08:37 | 00:07:56 | 00:06:92 | 00:05:62 |
| Redmi Note 8 Pro | 2019 | Android 10 Chrome 84 | 00:18:37 | 00:12:02 | 00:11:30 | 00:19:08 |
| Acer Swift SF314-52 | 2018 | Windows 10 Firefox 79 | 00:11:57 | 00:10:36 | 00:11:50 | 00:13:32 |
| Honor 8 | 2016 | Android 10 Chrome | 00:14:50 | 00:27:15 | 00:18:14 | 00:22:58 |
| HP-PC ProBook 450 G6 | 2018 | Manjaro Chromium | 00:11:59 | 00:09:79 | 00:10:09 | 00:09:45 |
| ASUS H81M-D R2.0 | 2015 | Mint 20 Mozilla Firefox | 00:10:28 | 00:11:57 | 00:08:92 | 00:08:54 |
| MacBookPro 14.2 | 2017 | macOS 10.15 Safari | No webGL1.0 | No webGL1.0 | No webGL1.0 | No webGL1.0 |

Each device was tested by resetting and clearing the browser cache and using the anonymous navigation mode in order to have the most accurate test as possible. The loading times are reported according to the notation: *minutes:seconds:hundredths of a second*. Since most of the times the project compressed when Brotli and Web Assembly Streaming were active was faster, we produced a summary Table 3 that shows the final results obtained comparing this type of compilation when using WebGL1.0 and WebGL2.0.

**Table 3.** Synthetic comparison of the results.

| *Device* | **Year** | **Operative System / Browser Web** | *WebGL 2.0 Brotli WAS* | *WebGL 1.0 Brotli WAS* |
|---|---|---|---|---|
| Asus Zenfone 5 | 2018 | Android 9 Chrome 85 | 00:16:10 | 00:09:51 |
| Amazon Fire HD 8 | 2017 | Fire OS 6 Chrome 84 | 00:12:63 | 00:12:62 |
| Samsung Galaxy Tab E | 2015 | Android 4.4.4 Chrome 81 | No webGL2.0 | 01:01:57 |
| Huawei MediaPad M5 lite | 2019 | Android 8 Chrome 84 | 00:28:87 | 00:26:54 |
| Samsung Galaxy S7 | 2016 | Android 8 Chrome 84 | 00:06:07 | 00:05:70 |
| Samsung Galaxy S4 Active | 2013 | Android 5.01 Chrome 85.0.4 | No webGL2.0 | 00:15:65 |
| DESKTOP CUSTOM | 2014 | Windows 10 Chrome 84 | 00:01:65 | 00:01:25 |
| Asus GL502VM | 2018 | Windows 10 Chrome 84 | 00:01:82 | 00:01:82 |
| Hp prodesk 400 g1 | 2013 | Ubuntu 18.04 Firefox 79 | 00:02:23 | 00:01:98 |
| DESKTOP CUSTOM | 2012 | Windows 10 Firefox 79 | 00:07:31 | 00:05:62 |
| Redmi Note 8 Pro | 2019 | Android 10 Chrome 84 | 00:16:54 | 00:19:08 |
| Acer Swift SF314-52 | 2018 | Windows 10 Firefox 79 | 00:12:30 | 00:13:32 |
| Honor 8 | 2016 | Android 10 Chrome | 00:15:17 | 00:22:58 |
| HP-PC ProBook 450 G6 | 2018 | Manjaro Chromium | 00:09:01 | 00:09:45 |
| ASUS H81M-D R2.0 | 2015 | Mint 20 Mozilla Firefox | 00:10:13 | 00:08:54 |
| MacBookPro 14.2 | 2017 | macOS 10.15 Safari | No webGL2.0 | No webGL1.0 |

As it can be seen some very old and outdated devices such as Samsung Galaxy S4 are not compatible with WebGL2.0 while they are able to run the project with WebGL1.0.

The results obtained show that between the two versions WebGL1.0 is faster than WebGL2.0. Finally, in Table 4 the size of the projects in MB is reported. As it can be seen the exported project using WebGL1.0 technology, Brotli and WebAssembly Streaming occupies 8189 MB, proving to be the lightest and most efficient combination for WebGL projects.

**Table 4.** Size of the exported project

| *Project* | **Size in MB** |
|---|---|
| WebGL 1.0 Gzip | 9965 |
| WebGL 1.0 Gzip WAS | 9967 |
| WebGL 1.0 Brotli | 8189 |
| WebGL 1.0 Brotli WAS | 8189 |
| WebGL 2.0 Gzip | 10,454 |
| WebGL 2.0 Gzip WAS | 10,455 |
| WebGL 2.0 Brotli | 8560 |
| WebGL 2.0 Brotli WAS | 8564 |

## 5. The Demo Exercises

The goal of the paper is to present the creation of a new environment, whose features have been previously described, that lets the therapist take advantage of the paradigm of rapid prototyping, to generate therapeutic exercises in a very fast and optimized way. To get this point, we show here a couple of Demo exercises that help understand the huge potential and effectiveness of the method and the complexity of the technical structure behind it. The actual maturity of Virtual Reality has allowed us to implement a new and rich environment where the patient can find a set of tools to extend the available possible practices, exploiting a virtual world.

Demo A involves the patient, asking her/him to identify non-contextual objects within a room. The objects that have been chosen, shown in Figure 3, are simple and unequivocal, so as to be sure that the error is due to a difficulty inherent in the subject rather than due to the type of stimulus to be recognized or individual variables. The objective of this activity is to train the executive functions of the subject and more particularly his ability to categorize. In fact, the objects placed inside the room must first be recognized by the subject and secondly defined as belonging to the category "typical objects present in the bedroom" or not. Demo B, instead, requires the patient to remember a list, made up of a variable number of objects positioned in an unconventional way in a room. The two demos have several aspects in common, in fact, both follow a very precise structure:

- explanation of the task that the user has to perform, both in a textual and auditory way.
- objects randomly appear in the room during the execution of the exercises, as shown in Figure 4; if the user correctly clicks on one of them, there is a positive audio reinforcement and the object rotates and disappears. Whereas only a negative audio reinforcement is returned if the object is in the right context, which means a user failure.
- feedback on the results is expressed with some stars and based on the ratio between the total number of clicks and the number of correct objects found. The score is expressed in such a way so that the patient does not feel frustrated even if she/he obtained a bad score. Moreover, phrases have been added to encourage the patient to keep on doing exercises in order to improve.

In addition, input variables and output variables have been associated with both of them: the input variables are parameters that can be set by the specialist, in order to customize the activity according to the skills and needs of their patients.

The output variables, instead, represent those parameters that are traced during the execution of the activity and that allow the therapist to monitor the rehabilitation path carried out by each patient.

Although they both share a common activity structure, the two demos present significant differences from a technical and clinical point of view.

Demo A is not so structured from the point of view of programming, as the scenario remains fixed, whereas in Demo B, the spatial exploration has been implemented and made possible by the use of directional arrows (the exploration is only permitted on the horizontal plane and not on the vertical one).

However, this technical difference also has a resonance on the clinical level because in the second case the user has to apply particular skills, such as visual-spatial orientation and eye-manual coordination. On the clinical level, finally, we choose to train different functions simultaneously. While in Demo A the patient primarily exercises categorization, in the other case we aim to train both the subject's mnemonic skills (working memory) and cognitive functions, with particular attention to cognitive flexibility.

The point is not only to remember what objects can be found inside a room, but also to be able to recognize them in an unusual perspective and position. We have chosen to insert a "HELP" key in the training screen: so the subject will be able to listen again to what objects she/he has to find inside the room. She/he can press the button and access the help menu every time it is thought to be necessary.

However, these kinds of actions are recorded and taken into account when the final score is assigned. The target objects found, the time to complete the activity and the errors made are all traced and stored. These last output variables are traced in Demo A as well. In this way, during the monitoring of the activities, the operator can distinguish if the subject mainly has a working memory deficit or if she/he presents difficulties in both areas.

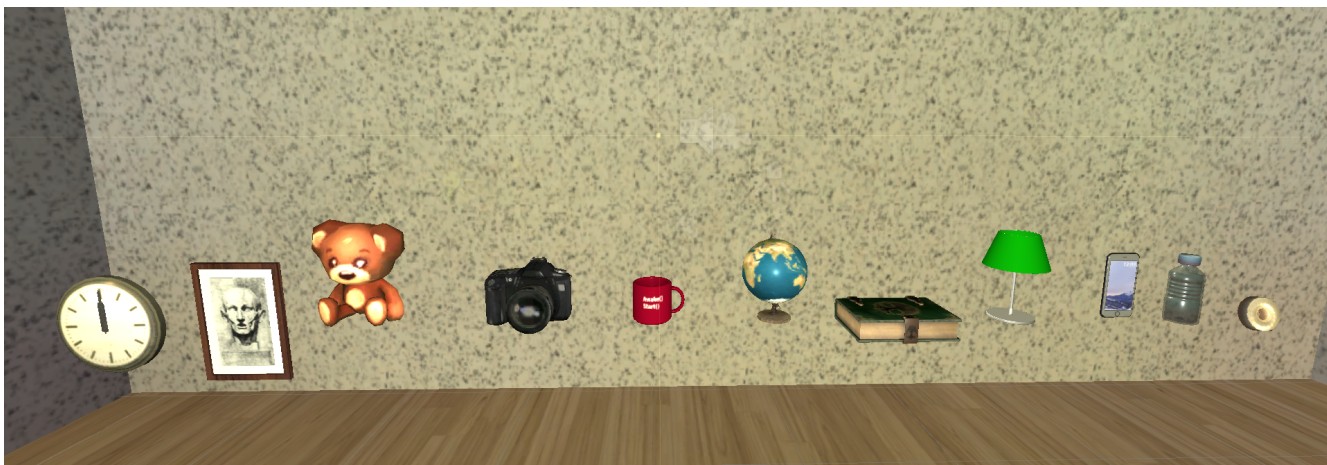

**Figure 3.** Demo A example with objects that will be randomly arranged in the room.

The exercises have a maximum duration, after that, if the user did not find all the non-contextual objects, she/he will receive a negative score.

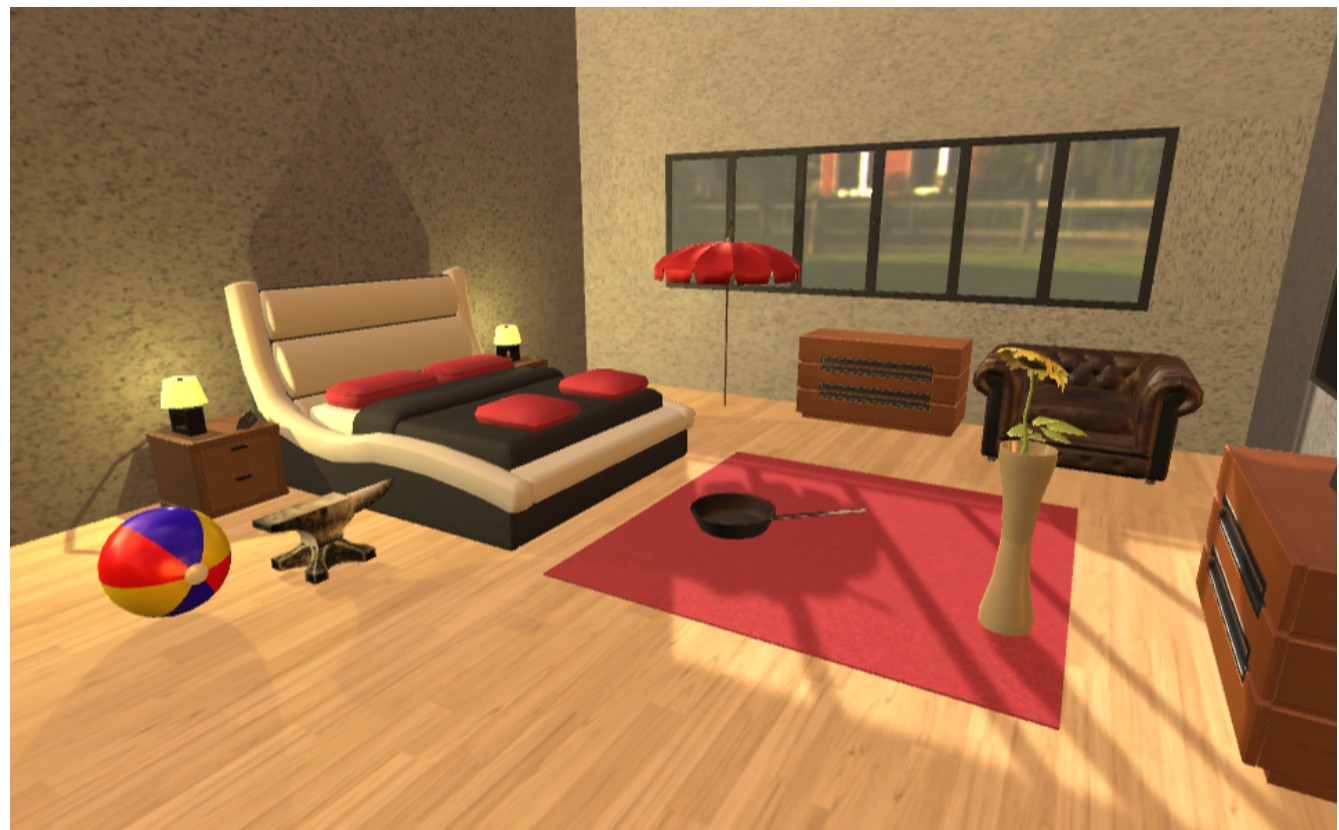

**Figure 4.** Demo A: sample exercise

It is possible to add additional sentences linked to audio tracks. The text part will be displayed on screen while the corresponding audio will be played and, at the end of the audio, the program will automatically pass to the next sentence, loading its audio track.

As soon as the explanation of the exercise is finished, another program is launched for randomly visualize the objects in the room. The program selects a series of coordinates inside the room and positions the objects one by one in the selected coordinates, taking them from the set of objects previously chosen by the doctor and checking that there are no overlapping objects or empty areas of the scene. This part has been developed to prevent the patient from remembering a certain sequence of areas where to click, trying to stimulate his ability to promptly recognize which objects do not belong to the shown room context.

Each non-contextual object is linked to the scripts that allow the rotation, its disappearance and the expression of positive reinforcement, while in the case of non-contextual objects, it has been assigned a negative reinforcement.

The rotation script, as soon as the mouse click on the object is detected, rotates the object of 180 degrees, emitting a sound that expresses the correct execution of the exercise. Then, if the object is not contextual, the script that makes the positive reinforcement and removes the object from the scene will be activated. In this case, we simplify as much as possible the understanding of the exercise by the patient, since the script removes from the scene the elements already discovered and facilitates the identification of the remaining ones.

## 6. Conclusions and Future Works

This paper describes our recent activity devoted to the improvement of tele-rehabilitation exercises for the patient affected by various types of impairments by a Virtual Reality environment. It is based on an efficient mix of Blender and Unity3D components, which let therapists use a rapid prototyping environment.

The described Demo exercises consent to the training of cognitive functions such as memory, attention and executive function. The therapist can customize the difficulty level of the activities according to the patient's abilities. Moreover, the patient's experience can be enriched by adding familiar virtual scene objects that make exercise effects more positive.

The virtual environment empowers the NU!Reha Service with a series of rehabilitation exercises that can be rapidly deployed and customized according to the patient's rehabilitative needs in a way to increase efficacy and motivation.

A careful and in-depth study was carried out to optimize the size of the applications developed in Unity3D so that these exercises can be run with the best performance and graphic quality on any device available today. This feature has got much importance because it permits the patient to use her/his own device; this makes the execution of the exercises cost-effective and friendly. Our future work in this area intends to deepen the data analysis and consolidate obtained results. It is very important for our work to go on collaborating with the team of rehabilitation experts to explore new ways to use technologies and help patients' recovery.

Our working group is already applying for trials to evidence the clinical efficacy of NU!Reha service seen as a general approach to tele-rehabilitation, as well as specific activities proposed in the platform. An evident advantage of the platform is the possibility to custom difficulty level and the features of therapeutic treatment by the therapist. A preliminary single-case study is very promising and this fact encourages our team to go further in the development path.

**Author Contributions:** Conceptualization, M.F., R.M., J.C. and O.G.; data curation, D.P.; formal analysis, M.F., R.M., J.C.; investigation, D.P., M.F., M.S., R.M., J.C. and O.G.; methodology, M.F., R.M. and O.G.; software, D.P., M.F.; supervision, R.M., O.G.; validation, D.P., M.S., M.F. and J.C.; visualization, M.S.; writing—original draft, D.P., M.F., M.S., R.M., J.C. and O.G.; writing—review and editing, D.P., M.F., M.S., R.M., J.C. and O.G. All authors have read and agreed to the published version of the manuscript.

**Funding:** This research received no external funding.

**Data Availability Statement:** Not applicable.

**Conflicts of Interest:** The authors declare no conflict of interest.

## Abbreviations

The following abbreviations are used in this manuscript:

| | |
|---|---|
| API | Application Programming Interface |
| AT | Assistive Technology |
| CNS | Central Nervous System |
| CPU | Central Processing Unit |
| GPU | Graphic Processing Unit |
| HTTP | Hyper Text Transfer Protocol |
| ICF | International Classification of Functioning, Disability and Health |
| I/O | Input/Output |
| JSON | JavaScript Object Notation |
| OS | Operating System |
| RAD | Rapid Application Development |
| RAM | Random Access Memory |
| REST | Representational State Transfer |
| RGI | Realtime Global Illumination |
| UV | Represent the u,v graphic coordinates |
| VR | Virtual Reality |
| VRAM | Video Random Access Memory |

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
