# Peer review of "Rapid Prototyping of Virtual Reality Cognitive Exercises in a Tele-Rehabilitation Context"

_electronics, doi:10.3390/electronics10040457_

Round 1

Reviewer 1 Report

Dear author, I've read your manuscript with much interest and I applaud you for the important topic you tackled. This manuscript describes the development of rapid prototyping of tele-rehabilitation VR platforms, specifically NU!Reha. The implementation of a Unity technique is described and tested on 17 different devices with good results. While I like this work, I think that the manuscript can be improved on a couple of main issues: reduce and narrow the introduction, discuss how neuro-psychological requirements have been included in the development of cognitive activities.

Introduction sections: too many aspects are presented in the introduction sections and it is easier to get carried away and loose the main point. I suggest authors to consider reducing and condense introduction, and focus only on the main aspects.

Cognitive tasks: it is unclear how neuro-cognitive requirements have been considered in the development of cognitive tasks. It is great that therapists can choose from a variety of cognitive tasks, but it is unclear whether the list of activities include tasks that are useful and essentials for therapists. In other words, does the software allow therapists to do what they want to do or allow them to design activities with little relevance for the various neuro-cognitive diseases? I suggest authors to carefully describe how cognitive tasks have been selected, what principles have been considered in the development, and how therapists can actually benefit from those tasks.

Lastly, I suggest to play down a little the fact that the developed tele-rehabilitation help recovery from injury/disease. Unless there is some evidence (not presented in the manuscript) on the effectiveness of this software, you cannot say with certainty that it will help patients. I suggest saying that the software 'may' help patients.

Author Response

We are deeply grateful for your words of appreciation and valuable advice. In the updated version, where the modified parts are highlighted in yellow, we have taken your valuable suggestions into account.

We reduced the Introduction, and pointed out how cognitive aspects affect the therapist's choice of exercises, and we modified the part about the effectiveness of the system.

Reviewer 2 Report

The article presents a platform for rapid prototyping, using Unity 3D, of a VR environment for cognitive exercises in a therapeutic context. It is however not clear what was already available in the Nu!Reha system, that was published about 10 years ago (and commercialized), and what is new in this paper. I rechecked parts of the paper to find this and highlight the contribution, but I’m not able to find that from a research point of view. Even if it is an “engineering” paper, it still is a scientific article that needs a clear contribution.

The domain of telerehabilitation is certainly very important nowadays, and sharing experience with the realization of the system can help colleagues. However, in its current version, the article is not yet polished enough regarding its scope and writing. I would advise the authors to rework the article considerably.

In general, the article misses a focus. It elaborately introduces the context of telerehabilitation from a medical perspective and related technical needs, but the paper’s real topic is hard to guess…based on the abstract the contribution is the platform. Or is it the realization of rapid prototyping? In the beginning of the paper a lot of attention is given to rapid prototyping, but further on it is hardly clarified how that is realized. The structure of the paper should reflect better what is context (e.g. intro and related work) and what is the new contribution. Some sections, such as 3: Relevance of VR cognitive exercises, are a bit isolated, not belonging to context nor to the core of the paper. Looking back at the paper after reading completely reveals that more than half of the paper is context, so the contribution that is put forward should be more central.

The introduction elaborates extensively on the need of cognitive exercises and on telerehabilitation. To my honest opinion, the paper’s quality would increase if the message on the aim of the system is given more concise. There are many factors determining the success of telerehabilitation, and it is not correct to state that hardware is the most important one.

Line 54: it is not clear what is meant by “device” in this context; a smartphone, training devices?

It is very strange to use the term BYOD in this context. What is presented is a web application that works on several platforms, so a multidevice or multiplatform application. BYOD is used to denote the fact that several users enter the same physical space with their own device, and interact with each other and with the environment.

The related work section lacks structure, and is way too long. It introduces a bit of everything, and e.g. the details on cognitive exercises are not relevant if this paper is a technical paper / engineering paper. Just introduce the exercises in one short paragraph. Same for telerehabilitation. Make sure you start the section with _what_ rapid prototyping is rather than some general sentences on that topic. Rethink the flow and story in this related work section, and restrict the description to related work that is directly related to the contribution of this paper (so technical).

Line 141: A ref from 2009 by a physiotherapist is not the best source for categorising technology for telerehab; a lot has changed since then, and also networking technology is very important.

In the middle of the related work section, line 148, a description of Nu!reha is given. It is too extensive for the purpose it has here, and could maybe be a separate section as the authors seem involved in that system too. Maybe in combination with the current section 4?

Line 152: too much self references!

Line 233: Only make a claim on improved effectiveness if you can add a ref to a study or demonstrate enhanced effectiveness in this article!

Section 4 on architecture of Nu!reha: as the system is not new, I guess this architecture has been published yet? Then provide a reference. If not, make clear what is new in this architecture. Starting here, I got lost in the paper of what is “context” and what is the new “engineering” contribution that is put forward.

If section 5 is intended to by the main contribution, it is not very convincing. It is written in a descriptive way on technical details. It should be more clear what you want the reader to learn from this section; e.g. the test results can be interesting, but do you expect readers to be programmers / analysts looking for these details? Also, information on user tasks / scenarios is mixed with technical details. I would expect to read more explicitly on “rapid prototyping” in this section. Give enough explanations for the tables with results; they are hardly mentioned now.

I would advise to reconsider the structure of the final section on conclusions and future work. Also, the future work is not directly related to this paper, but is a general remark on research in this field. Be more precise.

The article needs thorough proofreading by a native English speaker. That might also result in a more uniform writing style. Currently, the first part of the paper suffers more from language issues than the second part. Several sentences are too long, with a lot of sub-sentences, and sometimes the structure is not correct what makes it hard to understand the exact meaning of the sentences.

Author Response

We are deeply grateful for your valuable advice. In the updated version, where the modified parts are highlighted in yellow, we have taken your valuable suggestions into account. 

We pointed out what is new in NU!Reha 

The Introduction has been rewritten and shortened.

We have clarified the innovative aspects of the NU!Reha platform and have gone into more detail about rapid prototyping.

The term BYOD has been removed and the related work section rewritten.

The reference of 2009 has been dropped.

NU!Reha description has been rewritten.

Self references have been reduced

The remark originally on line 333 has been removed.

Section 4 has been revised 

Section 5 has been revised to point out the relevant work carried out to find the optimal implementation of the various components of the system.

Conclusions and future work has been restructured.

The english has been carefully proofread by an english native speaker.

Reviewer 3 Report

The work presents a IoHT technique, the overall quality is acceptable. Here are some suggestions for authors.

1. The format is expected to be improved. For example, font size of the tables are different, some text cannot be read clearly.

2. the flow of the paper is not well organized, logic connection between paragraphs and sentences should be created. The language should be further polished.

Author Response

We are deeply grateful for your valuable advice. In the updated version, where the modified parts are highlighted in yellow, we have taken your valuable suggestions into account. 

The structure of the paper has been improved. 

The Tables have been revised.

The language has been improved.

Round 2

Reviewer 2 Report

The article presents a platform for rapid prototyping, using Unity 3D, of a VR environment for cognitive exercises in a therapeutic context. Compared to the first version of the paper, the article clarifies better wat was available in the Nu!Reha system, and how it was adapted.

The domain of telerehabilitation is very important nowadays, and sharing experience with the realization of the system can help colleagues.

Similar to the first version that was submitted, the article misses a focus, even if the authors took some suggestions by the reviewers into account.

It elaborately introduces the context of telerehabilitation from a medical perspective and related technical needs, but the paper’s real topic seems to be the adaptation of the platform, so more from an engineering perspective. In the new version rapid prototyping is more elaborately introduced in the related work, but this focus disappears a bit in the following sections.

The structure of the paper did improve a bit when considered in the sections themselves, let’s say the flow within the sections is more fluent to read.

In particular, the related work section is structured a bit better, but still introduces a bit of everything. This is another “symptom” : if the focus of the overall article was more clear then the most important topic for the related work was easier to select.

As asked in the previous review report, the authors better define _what_ rapid prototyping is and why they introduce the terminology for this article.

It is a better choice to group the description of Nu!reha in a section as it is currently done. It is still extensive and I am wondering to what extent e.g. the conceptual / architectural figure was already published previously.

The authors mention that sections 4 and 5 have been revised. There is indeed a yellow marked part in section 4. Section 5 has no marked changes, and when comparing the flow it does not seem to have changed considerably. Still section 4 and 5 together are key sections to highlight the contribution. In that sense, it is a bit disappointing that the article did not improve enough here, though the topic of the article and the general idea of the content on rapid prototyping for these VR exercises remain interesting!

The English writing improved, apart from some language issues spread throughout the paper (e.g. “family’s patient device”).

Author Response

Dear Reviewer,

we would like to thank you for the very precise and timely notes for the precise and timely notes that allowed us to better express our thoughts and significantly improve the quality of our article.

According to your recommendations we modified the introduction, related work, and section 5, emphasizing the inspiring principles of the paper and the most important results obtained with our work.

We wish to hilight that we have created a new environment that will enhance the possibilities and effectiveness of NU!Reha platform, and is also inspired by the principle of rapid prototyping that enables the easy and fast implementation of a homogeneous set of tele-rehabilitation exercises.